# Label-Free, Portable, and Color-Indicating Cholesteric Liquid Crystal Test Kit for Acute Myocardial Infarction by Spectral Analysis and Naked-Eye Observation

**DOI:** 10.3390/bios13010060

**Published:** 2022-12-30

**Authors:** Fu-Lun Chen, Li-Dan Shang, Yen-Chung Lin, Bo-Yen Chang, Yu-Cheng Hsiao

**Affiliations:** 1Department of Internal Medicine, Division of Infectious Diseases, Wan Fang Hospital, Taipei Medical University, No.111, Sec. 3, Xinglong Rd., Wenshan Dist., Taipei 11600, Taiwan; 2Department of Internal Medicine, School of Medicine, College of Medicine, Taipei Medical University, 250 Wuxing St., Taipei 11031, Taiwan; 3Department of Geography and Planning, University of Liverpool, Liverpool L69 3BX, UK; 4Department of Internal Medicine, Division of Nephrology, Taipei Medical University Hospital, 252 Wuxing St., Taipei 110, Taiwan; 5TMU Research Center of Urology and Kidney (TMU-RCUK), Taipei Medical University, Taipei 110, Taiwan; 6Department of Biomedical Engineering, Taipei Medical University, Taipei 11031, Taiwan; 7Graduate Institute of Biomedical Optomechatronics, College of Biomedical Engineering, Taipei 11031, Taiwan; 8International PhD Program for Biomedical Engineering, Taipei Medical University, Taipei 11031, Taiwan; 9Cell Physiology and Molecular Image Research Center, Wan Fang Hospital, Taipei Medical University, Taipei 11031, Taiwan

**Keywords:** acute myocardial infarction rapid test, cholesteric liquid crystal, cardiac troponin I (cTnI)

## Abstract

The early diagnosis of acute myocardial infarction is difficult in patients with nondiagnostic characteristics. Acute myocardial infarction with chest pain is associated with increased mortality. This study developed a portable test kit based on cholesteric liquid crystals (CLCs) for the rapid detection of AMI through eye observation at home. The test kit was established on dimethyloctadecyl[3-(trimethoxysilyl)propyl]ammonium chloride-coated substrates covered by a CLC-binding antibody. Cardiac troponin I (cTnI) is a major biomarker of myocardial cellular injury in human blood. The data showed that the concentration of cTnI was related to light transmittance in a positive way. The proposed CLC test kit can be operated with a smartphone; therefore, it has high potential for use as a point-of-care device for home testing. Moreover, the CLC test kit is an effective and innovative device for the rapid testing of acute myocardial infarction-related diseases through eye observation, spectrometer, or even smartphone applications.

## 1. Introduction

An early acute myocardial infarction (AMI) diagnosis can be difficult for patients with nondiagnostic features. Cardiac troponin I (cTnI) is the inhibitory subunit of the troponin complex and can inhibit the binding of actin and myosin; cTnI plays a major role in AMI [1,2,3]. The studies have shown that cTnI concentration increases rapidly during the early stage of AMI [4,5]. Furthermore, cTnI is highly specific to cardiomyocytes but not to skeletal muscles [6,7,8]. The cTnI has a sensitivity and specificity of 96% and 97% in AMI diagnosis, respectively [9,10,11]. In addition, cTnI serves as a major serum biomarker of injury for myocardial cellular, and it affords the advantage of clear diagnostic thresholds and rapid detection, which are crucial for the assessment of AMI patients with myocardial cell injury [12,13]. The cTnI concentrations are associated with the extent of the cardiac injury; the high cTnI concentrations (100 μg/mL) indicate myocardial infarction [13,14]. Therefore, quick cTnI quantification and detection are crucial for the assessment of cardiac injury, thus necessitating the development of a label-free, rapid, and low-cost cTnI detection bio-device. Several methods are available for measuring cTnI, including ELISA [15,16], fluorescence immunoassay [17,18,19], colorimetric [20], electrochemical [21], paramagnetic [22], and surface plasmon resonance [23] methods, etc. However, when compared to liquid crystal (LC) biosensors, these approaches are disadvantaged due to sophisticated equipment and time-consuming processes. Accordingly, developing a fast, label-free, sensitive, and low-cost device for detecting the cTnI level is imperative.

Recent studies have shown that LC biosensors can successfully diagnose diseases [24,25,26,27,28,29,30,31,32]. Biomolecules can redirect LC molecules, causing changes in the light signal’s intensity [33]. The redirected LC molecules cause a sensitive response to changes in the optical signal. The studies have integrated LC with microfluidic detection of bovine serum albumin [34,35]. Additionally, cholesteric liquid crystals (CLCs) have distinct viewing angles when compared to nematic LCs: bistability, Bragg reflection, and flexibility [36,37,38,39]. In 2015, the CLC high-sensitivity biological sensor (limit of detection = 1 fg/mL) was developed for detecting bovine serum albumin [40].

The present paper invented the color-specific CLC test kit for rapidly testing AMI through eye observation. The study used a pair of substrates coated with a cTnI antigen/peptide and the CLCs to perform an assay. The proposed CLC test kit is different from previous normal biosensors. The level of cTnI antigen/peptide could be observed and measured with the naked eye. The alignment layers of N,N-dimethyl-n-octadecyl-3-aminopropyltrimethoxysilyl chloride (DMOAP) were employed to sense the cTnI levels [41]. The schematic of the proposed CLC test kit is illustrated in Figure 1. According to our review of the literature, this is the paper to use color CLC biochips as a new medical approach to identify the pathological severity of AMI-related diseases through early detection of cTnI. The innovation of the study is that it presents the bio-device for rapid AMI testing at home with point-of-care.

## 2. Materials and Methods

The mechanism of the proposed novel CLC test kit is displayed in Figure 2. The test kit executes optical detection using cross-polarized microscopy through a spectrometer. The test kit estimates the concentration of cTnI in a sample according to the intensity of light signals. DMOAP—serving as the alignment layer in the sensor—has a long carbon chain that can align CLC molecules vertically. The nematic LC E7 and chiral dopant R5011 were used in this study’s experiment. To enable DMOAP to respond to cTnI, it is covered by the additional JP5-19H peptide as an antibody layer. Specific immune complexes crowd out and disrupt the arrangement of CLC molecules. The adsorption of an immune complex on a substrate engenders a bright light signal owing to the planar state of the CLC test kit (Figure 1, right panel). Because the light intensity is positively related to the cTnI concentration, the presence of cTnI can easily be recognized. Accordingly, the biosensor quantifies the light intensity signal passing through the CLC molecule, thus enabling the estimation of the level of cTnI in unknown samples.

Step 1. Coating the alignment layer:

Glass substrates were used in the experiments. The substrates were cleaned with deionized water (DI water) and vibrated ultrasonically twice in the container for 15 min in order to eliminate dust. Subsequently, wash with DI water (50 mL) for half a day, followed by DECON-90 with 2.5 mL. The substrates were rinsed with DI water and vibrated in the ultrasonic shaker for 15 min again. We chose the DMOAP as the alignment layer. DMOAP is C18H37, which arranges CLC molecules vertically (Figure 1). The cleaned glass was nicely stacked in the container and completely immersed in the DMOAP with 1 wt% Subsequently, a solution was agitated on an ultrasonic shaker for several minutes and rinsed with DI water to remove any excess DMOAP.

Step 2. Producing the immunoassay layer:

Single biomolecular layer: According to the detection principle for CLC test kits, a single biomolecular layer should be used to assess different concentrations of cTnI samples. As a result, different levels of cTnI were produced using DI water solvent. To obtain the LOD, we first employed a broad concentration range (i.e., 0.01, 0.1, 1, 10, and 100 g/μL) and lowered the concentration range gradually.

Preparing double immunoassay layers: A double-layer immunoassay was used to examine the effect of the brightness on the different levels of cTnI and peptides, measure optical intensity, and confirm the brightness-concentration association. We found the optimal peptide concentration for cTnI detection and measured the brightness after validating the link between peptide and cTnI levels. The required preparation steps are outlined below.

The peptide drops (0.5 µL) at different concentrations on the alignment layer. After the lower substrate had dried, cTnI with 40 μL was dropped on the first peptide layer and covered with another clean substrate to ensure that cTnI was equally distributed on the layer of peptide. We allowed 1 h for the immune reaction between the applied cTnI and the peptide. Next, the upper glass substrate was removed, and then the lower glass substrate was dried.

Step 3. Making the CLC layer:

The two thin glass substrates with the alignment DMOAP coating were used to make a CLC cell. The CLC E7 and the chiral dopant R5011 were filled between the two basic glass substrates. Subsequently, CLCs were distributed in the cell uniformly through capillary action. Finally, we seal four edges of the substrate using AB glue.

## 3. Results

In order to display the characteristics of the CLC test kit that can be observed by the naked eye and smartphones. Figure 3 depicts polarized optical images captured by the proposed CLC test kit—operated by a smartphone—under cross-polarization. The optical texture of the CLC test kit under various concentrations of cTnI is displayed in Figure 3. We observed that the optical brightness increased with the cTnI concentration. Moreover, we observed that the CLC molecules were randomly in a focal conic (FC) state around the two substrates. When cTnI was immobilized on the coated substrate, the CLC molecules were noted to be completely in a planar (P) state. Light passing through the CLC material was scattered by some of the CLC molecules in the FC state. However, when cTnI was immobilized, the capacity of DMOAP to maintain the vertical alignment of the molecules was reduced (Figure 2). As the cTnI concentration increased, the CLC molecules switched from the FC state to the P state. Finally, the optical intensity of the CLC molecules in the P state increased (Figure 3). The optical texture of four CLC test kits under various concentrations of BSA is illustrated in Figure 3. It is worth noting that the chiral dopant material R5011 used in this study is temperature-independent [20]. Therefore, CLC test kits can be used conveniently and consistently at a wide range of room temperatures in different countries. In addition to using POM, CLC can be directly observed by naked eyes from the change in color [40]. In order to be able to more accurately determine the concentration of the cTnI, we evaluated light intensity using the spectrometer with a fiber-based design and microscopy with cross-polarizers, a technique known as micro-reflective fiberoptic spectrophotometry (MRFS). We used a white LED light source. After the light had penetrated through the microscopic system, we measured cTnI concentrations on the basis of the intensity of the light (Figure 4). Polarized optical images of the CLC test kit were captured after the cTnI peptide interacted with cTnI of various concentrations under cross-polarization. The brightness level increased with the cTnI concentration under cross-polarization. The transmission spectra of the CLC test kit under 0–1 mg/mL of cTnI are presented in Figure 4. Results showed an LOD of 1 ng/mL for cTnI. Furthermore, it was revealed that the concentrations of cTnI were significantly associated with the intensities of the CLC test kit. We determined the detection mechanism and performance of the CLC test kit by using MRFS spectra. The transmittance spectra differed for the various concentrations of cTnI (0 μg/mL, 1 ng/mL, 1 μg/mL, and 1 mg/mL). We used a chromophore to measure the transmittance of the CLC test kit, with the minimum transmittance being observed at 500 nm due to the selective reflection of the CLCs (Figure 4). We captured the minimum intensity of light penetrating different concentrations of cTnI were quantified at a wavelength of 500 nm (Figure 5). The results show that the light penetrating the system at 500 nm is positively linearly related to the concentration of cTnI. It is worth noting that the penetration of the CLC chip will decrease after one hour, so the observation time must be completed within one hour.

To bind the cTnI, we add another peptide layer. The intensity of transmitted light was then assessed at various peptide and cTnI concentrations, as shown in Figure 6, where the *x*-axis represents the level of peptide or anti-cTnI and the *y*-axis represents the optical intensity. The data show that the optimal peptide concentration for cTnI detection was 10 μg/mL. When the peptide concentration was excessively high at 20 μg/mL and 30 μg/mL, the peptide severely interfered with the CLC molecule. In addition, even when the cTnI concentration was zero, the transmittance was high. By contrast, when the peptide or anti-cTnI concentration was excessively lower than 20 μg/mL, the reorientation of the CLCs was not affected by cTnI. Accordingly, we could not detect the different concentrations of cTnI owing to the lack of a notable difference in brightness. When samples containing the peptide were filled between the double immunoassay layers, the disturbance power of the LC molecules increased. The LOD of cTnI was determined to be lower in the double immune layer than that in the single layer. Accordingly, the LC biosensor for rapid myocardial infarction testing has favorable selectivity between cTnI and biomolecules.

After identifying the optimal peptide concentration with 10 μg/mL prepared for cTnI detection, we employed it to assess the efficiency of the proposed CLC test kit in detecting different amounts of cTnI with 0–50 μg/mL. In addition, the test shows that the data was applied to construct a calibration curve, as indicated in Figure 7. The data revealed that the cTnI concentrations were positively and linearly correlated with the optical intensity. Hence, the results indicate that cTnI concentrations in samples can be easily estimated using linear interpolation, as long as the concentration is within LOD. Furthermore, we established that the observed cTnI was 0–50 μg/mL (optimal peptide concentration: 10 μg/mL), with a 50 μg/mL exhibiting the brightness gain. Note that the reflection band of traditional CLC is very temperature-sensitive, but according to different chiral dopants, this conclusion has changed. Figure 8 shows the E7 doping with R5011, which is quite insensitive to temperature. In addition, the selectivity tests in cTnI are also shown in Figure 8.

## 4. Conclusions

This study is the first to present a CLC test kit for rapid AMI testing. The CLC test kit can detect monolayer cTnI samples and shows a LOD of 1 ng/mL for cTnI. Under crossed polarized microscopy, the cTnI level was positively linked with transmitted intensity. The photon has the maximum cyan reflectance at a wavelength of approximately 500 nm, as revealed by a spectrometer. We also determined that the optimal peptide concentration for binding cTnI between immunoassay layers was 10 μg/mL. In addition, we observed a positive relation between the concentration of cTnI and the transmittance of light. The LOD of cTnI immunoassay double layers, on the other hand, was less than 10 μg/mL. Furthermore, when observed using a smartphone camera, the optical intensity showed the same clear pattern of positive correlation as that observed with the naked eye. These results corroborate the potential of successfully testing for AMI by measuring the concentrations of cTnI. Accordingly, the proposed test kit has potential for use as a point-of-care test kit operated with a smartphone for home or clinical testing of AMI. Nematic liquid crystals provide an efficient method for rapid AMI testing using cross-polarized microscopes, smartphones, spectrometers, or the naked eye.

## Figures and Tables

**Figure 1 biosensors-13-00060-f001:**
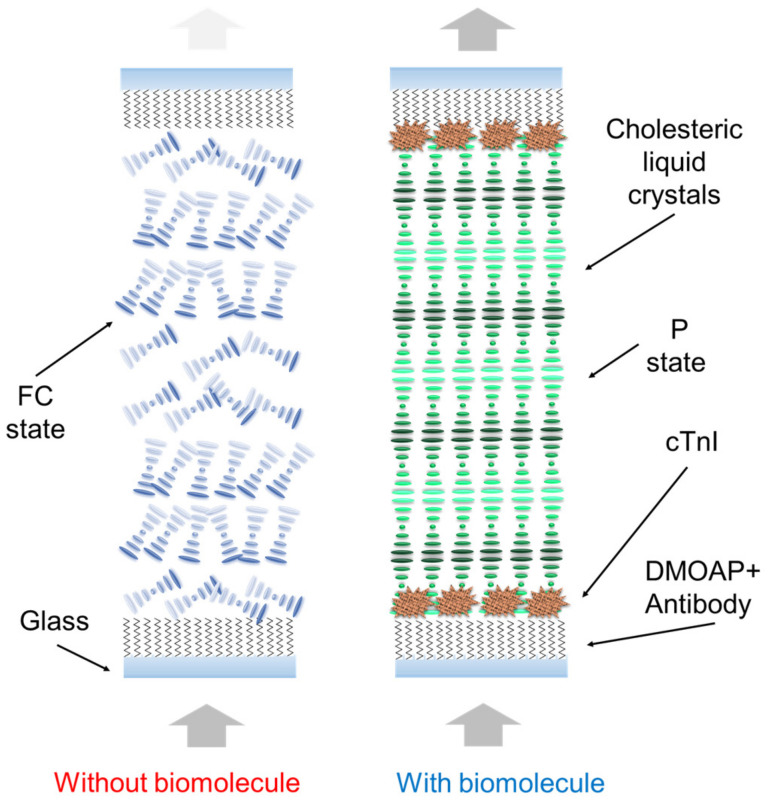
Schematic of the CLC test kit for rapid AMI testing. DMOAP enables CLC molecules to be vertically aligned. Schematic of the cTnI and peptide immunocomplex on the glass substrate.

**Figure 2 biosensors-13-00060-f002:**
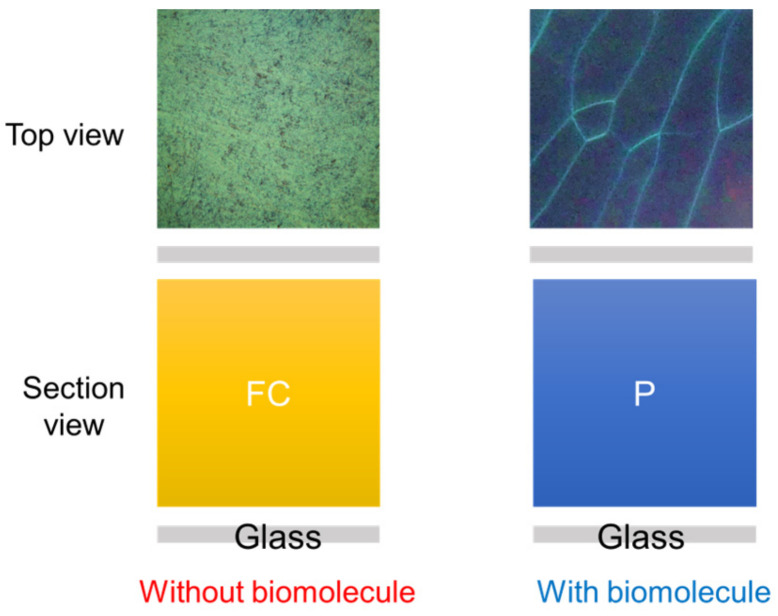
In the presence of abundant biomolecules on the DMOAP substrate, the configuration changes from a predominant reflection mode to a predominant transmission mode.

**Figure 3 biosensors-13-00060-f003:**
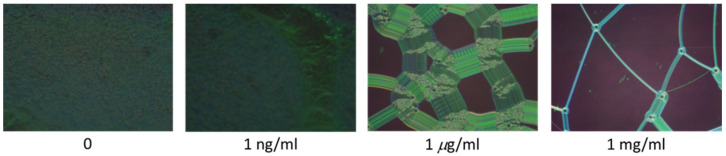
Polarized optical microscopic images captured by the CLC test kit operated by a smartphone under various concentrations of cTnI (0–1 mg/mL) immobilized on DMOAP-coated glass.

**Figure 4 biosensors-13-00060-f004:**
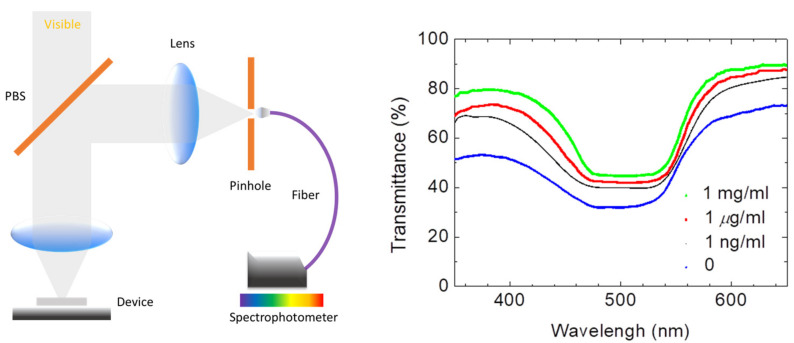
The schematic diagram of MRFS system and the spectra of the CLC test kit under 0–1 mg/mL of cTnI.

**Figure 5 biosensors-13-00060-f005:**
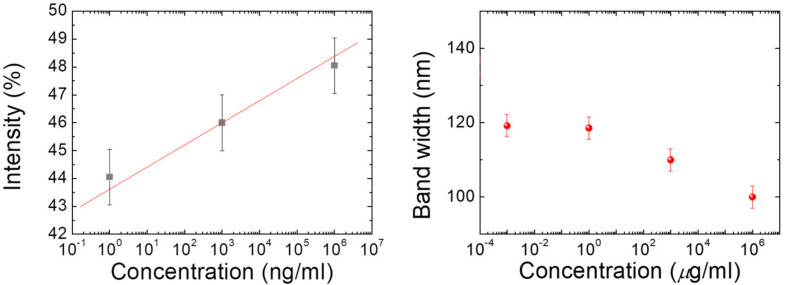
Linear correlation (determination coefficient R2 ≥ 0.93) obtained using 0–1 mg/mL of cTnI.

**Figure 6 biosensors-13-00060-f006:**
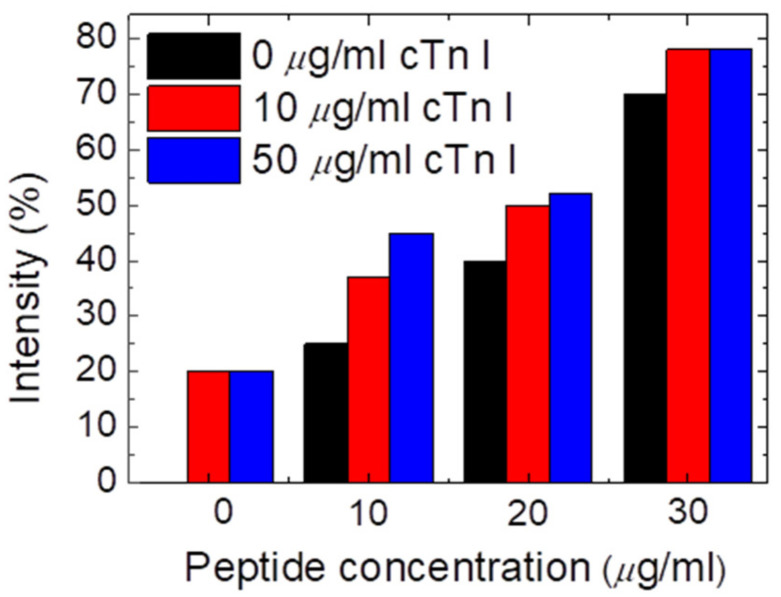
Intensity of an immunoassay CLC chip immobilized with peptide (concentration range from 0 to 30 µg/mL) and cTnI (concentration range from 0 to 50 µg/mL).

**Figure 7 biosensors-13-00060-f007:**
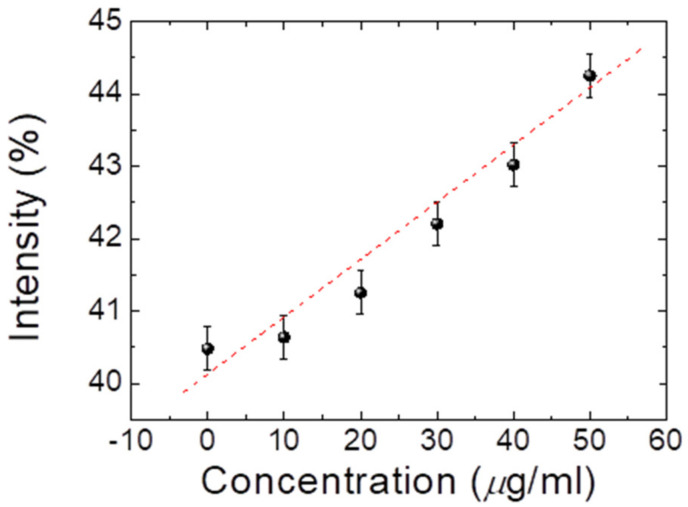
Correlation of light intensity with various cTnI concentrations in CLC biosensing chips with 10 μg/mL of peptide.

**Figure 8 biosensors-13-00060-f008:**
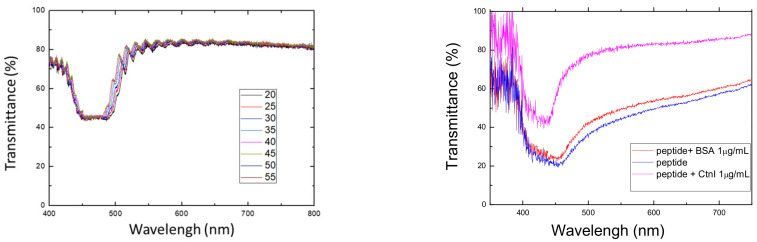
The spectra of the CLC test kit under different temperatures (left) and peptides with different biomolecules (right).

## Data Availability

Not applicable.

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
