# Peer review of "Label-Free, Portable, and Color-Indicating Cholesteric Liquid Crystal Test Kit for Acute Myocardial Infarction by Spectral Analysis and Naked-Eye Observation"

_biosensors, 2022, doi:10.3390/bios13010060_

Round 1
Reviewer 1 Report
Although the attempt of rapid self-detection for AMI at home through eye observation is very imaginative and sounds very promising, the actual content of the manuscript is not quite correspond to his ambitious purpose and striking title. See the attachment for the details.

Reviewer 2 Report
This manuscript presents the utilization of cholesteric liquid crystals to detect the cardiac troponin I (cTnI) that plays a major role in the acute myocardial infarction (AMI). By leveraging the intriguing feature of LCs in which the molecular configuration is strongly influenced by bounding surfaces, they designed the label-free, rapid, and low-cost AMI sensor that can resolve issues in previous methods. In addition, Authors adopt JP5-19H peptide to provide the selectivity into their sensor. Overall, the manuscript is readable and can provide another insight into the design and application of LC-based bio sensor. However, I would like to request Authors to address the following critical points by which the quality of work will be dramatically improved.
1. I found few overstatements. For example, Authors claims that “…this is the first paper to use the color CLC biochips as a new medical approach.”. I am aware of many examples that use CLCs for the detection of biomolecules. Previous works may not specify the application of their systems, but it is clear that they can be used in the medical field, too.
2. Despite the employment of CLCs, Authors used the “transmitted light” for the detection and quantification of trigger. I believe Authors think the one of their significances is the capability of “naked-eye detection” because the ability is specified even in the title of manuscript. If they really do, for the analysis they should use “reflectance or reflection spectrum” that is the distinct feature of CLCs enabling “naked-eye observation” even under the ambient light. In this reason, I don’t see any rational reason to use CLCs in their system. Other LC phases (e.g., nematic) should show the changes in transmittance due to anchoring transition.
3. In the absence of CTnI, why do Authors think the LC layer contain the planar region between focal conic regions (Figure 1 left)? I find neither theoretical reason nor experimental data for the presence of intermediate planar region.
4. I cannot find the thickness of LC layer used in the experiments. Because the amplitude of scattering in FC state and of transmittance in P state are dependent on the thickness of LC layer, it should be specified and precisely controlled.
5. Authors state that because the property of chiral dopant is temperature independent, their sensors can be used consistently at a wide range room temperature. This is incorrect. First of all, the helical twisting power is not determined solely by the chiral dopant but by the interaction between LC and dopants. In addition, because all characteristic of LCs (retardation, elasticity, anchoring strength) are strongly dependent on temperature, the results provided in the manuscript cannot be temperature independent. Additionally, the temperature should be specified and I wonder Authors perform the experiment at constant temperature.
6. The approach of additional peptide layer is great because it can provide the selectivity to target that is one of most crucial factors in the design of sensor. I understand that the used peptide is highly specific to cTnI. Because Authors used the peptide and DMOAP together, however, they can show different behavior. Also, their interactions with LCs may exacerbate the selectivity of peptide in this system. I don’t find any demonstration to show their sensor is indeed specific to target. Therefore, control experiments are required using other bio-molecules that are similar to the target.
If Authors address the suggestions and applied their answers/additional experiments into the manuscript, I would be happy to recommend the publication.
Reviewer 3 Report
The article depicts a novel method to myocardial Infarction using the cholesteric liquid crystals. The LC molecular configuration upon the biomarker and the antibody was different and the concentration of the biomarker was closed related to the reflection intensity of the CLC. The idea was interesting and the experimental design was smart. I would suggest its publication after addressing the following comments.
(1) In line 138, I’m curious about the conclusion on the 3D layered texture from the 2D POM image. It is better to provide more details.
(2) The CLC pitch and thus the reflection bandwidth and intensity was dependent on temperature. I’m also curious about the conclusion that the test was temperature independent.
(3) In Figure 5, the wavelength range where the intensity was integrated should be provided.
(4) The repeatability and reliability was crucial in medical tests. Thus, it is better to provide such experimental data. In addition, I’m wondering whether the intensity and bandwidth was stable with time. For example, the difference between the test data measured in 5 minutes and that measure in 30 minutes.
(5) The sensitivity of the method for practical application should be discussed.
(6) I’m also confused by the “Naked-eye Observation” that the observation was carried on using the POM and the spectrometer.
Reviewer 4 Report
The paper presents the innovative cholesteric liquid crystal biosensor for acute myocardial infarction visualization and possible analysis. The midethyloctadecylaminopropyltrimethoxysilyl chloride (DMOAP) was applied as vertical alignment (VA) layer for cholesteric liquid crystal (CLC) mixture, obtained by doping of nematic liquid crystal E7 with chiral dopant R5011 to cholesteric selective reflection at ~500nm (cyan-green). For vertical alignment CLC helix is along the substrate and gives low reflection signal observed normal to the substrate.
Coating of DMOAP with JP5-19H peptide - an antobody for cardiac troponin I (cTnI) - modefied the conditions of alignment layer from vertical to planar alignment. For planar alignment case CLC helix is normal to the substrate and gives high reflection signal observed normal to the substrate.
Once cTnI interacts to the antibody on the surface alignment conditions changes, causing scattering that is observed in between crossed polarizers. When the concentration of the peptide is optimal than the light scattering signal is linear to the concentration of the cTnI within the required range of detection 0-50 ug/mL. The optimal peptide concentration was determine.
The idea is great and the proof-of-concept demonstrated. However experimental implementation is very weak and can be significantly improved.
1) Nothing is said on the thickness of the CLC layer. The light signal intensity depends on the LC layer signal.
2) How do you determine 100% level of signal Fig.5 and Fig.6?
3) Fig.1a (left) is physically wrong. LC is surface stabilized. Thus once alignment conditions does not provides uniform alignment defects are within the whole volume of the liquid crystal. P state in the middle can not happen for CLC based on E7 doped with chiral dopant.
4) Colesteric liquid crystal has properties of selective reflection of circular polarized light. In your experiment circular polarization property is not used. The observation is perfromed between crossed linear polarizers. Thus simple nematic E7 might give even better scattering.
5) The results section is of low scientific quality. The obtained graphs are hypish, as the scattering signal determined via transmission really depends on the thickness of the LC layer, light source wavelength, polarization degree of the polarizers, temperature and etc. The result is not scientifically grounded. Since the all optical scheme is wrong. All observations and measurements of CLC should be performed with circular polarizers!
The CLC property of selective reflection is not used for measurements. There is no issue to use cholesteric LC for linear polarizers as nematic might give similar scattering.
6) line 189-190: '... observed cTnI was 0-50 g/mL ...' but seems that it should be ug/mL
7) line 209: '10 g/mL' - should be ug/mL
8) line 162-163: 'We used a chromophore to measure transmittance of the CLC biosensor...' - What do you mean? The peak you observed is due to selective reflection of the CLC mixture. It is not the absorption peak!
The most valuable information from all the paper is the idea and that the optimal peptide concentration is 10um/mL. The optical experiment part is of poor quality, while Fig.1a is physically incorrect.
However this is a good piece of research and I am convinced that it should be published. However major revision is needed.
Round 2
Reviewer 2 Report
Authors have addressed most of my comments and suggestions in a satisfactory way.
However, Authors' answers (and results in the response letter) for comment 5 & 6 are not applied properly in the revised manuscript.
I suggest Authors add the relevant results into either main manuscript or supporting information.
Reviewer 4 Report
line 109: '... employed a broad concentration range (i.e., 0.01, 0.1, 1, 10, and 100 g/mL)' - 100g/ml??? check the dimensions guess it should be ug/ml
line 167-168: '...(Figure 4). Note that the part of the reflected light of chips can be known by T+R=1.' - Better remove that statement. Once there are absorption or scattering in the sample T+R=1 is not valid. Better show the actual optical scheme how you measured transmission. Fig.4a is for reflection measurement.
The paper shows interesting preliminary results that demonstrate the proof-of-concept. The obvious physical flaws have been corrected. The authors are not specialists in cholesteric liquid crystals alignment. But it is clear that modifying the measurement procedure the results part can be significantly improved.
Still I was able to grab interesting points out of the paper. Accept with minor revisions.
